# Design and Control of a 1-DOF Robotic Lower-Limb System Driven by Novel Single Pneumatic Artificial Muscle

**Tsung-Chin Tsai and Mao-Hsiung Chiang ***

Department of Engineering Science and Ocean Engineering, National Taiwan University, No.1, Sec.4, Roosevelt Rd., Taipei 10617, Taiwan; d03525005@ntu.edu.tw

\* Correspondence: mhchiang@ntu.edu.tw; Tel.: +886-2-3366-3730

**Abstract:** This study determines the practicality and feasibility of the application of pneumatic artificial muscles (PAMs) in a pneumatic therapy robotic system. The novel mechanism consists of a single actuated pneumatic artificial muscle (single-PAM) robotic lower limb that is driven by only one PAM combined with a torsion spring. Unlike most of previous studies, which used dual-actuated pneumatic artificial muscles (dual-PAMs) to drive joints, this design aims to develop a novel single-PAM for a one degree-of-freedom (1-DOF) robotic lower-limb system with the advantage of a mechanism for developing a multi-axial therapy robotic system. The lower limb robotic assisting system uses the stretching/contraction characteristics of a single-PAM and the torsion spring designed by the mechanism to realize joint position control. The joint is driven by a single-PAM controlled by a proportional pressure valve, a designed 1-DOF lower-limb robotic system, and an experimental prototype system similar to human lower limbs are established. However, the non-linear behavior, high hysteresis, low damping and time-variant characteristics for a PAM with a torsion spring still limits its controllability. In order to control the system, a fuzzy sliding mode controller (FSMC) is used to control the path tracking for the PAM for the first time. This control method prevents approximation errors, disturbances, un-modeled dynamics and ensures positioning performance for the whole system. Consequently, from the various experimental results, the control response designed by the joint torsion spring mechanism can also obtain the control response like the design of the double-PAMs mechanism, which proves that the innovative single-PAM with torsion spring mechanism design in this study can reduce the size of the overall aid mechanism and reduce the manufacturing cost, can also improve the portability and convenience required for the wearable accessory, and is more suitable for the portable rehabilitation aid system architecture.

**Keywords:** pneumatic artificial muscle; torsion spring; fuzzy sliding mode controller; path tracking control

## 1. Introduction

Rehabilitation robots have increasingly become popular in the field of robotics, since they can not only provide a support for patients with impaired limbs or elders facing difficulties doing activities of daily living on their own, but also augment the power of able-bodied people. Of all the actuators, pneumatic artificial muscles (PAMs) may be the most promising due to their inherent compliance, which guarantees safe interactions between the operator and the device. In addition, high power-to-weight ratio and lightness are also ideal features for the applications of human-friendly devices. However, the non-linearity is the drawback that is required to mitigate for accurate control.

Recently, many countries have studied a new type of PAM actuator for industrial applications and robotics. In the 1960s, this was invented by an American doctor, Joseph L. McKibben [1].

In addition to the movement of muscles similar to biological muscles, artificial pneumatic muscles still have unique advantages in terms of force performance, appearance and function, such as: 1. Simple structure, light weight, easy to miniaturize. 2. Flexible, they will not hurt the object. 3. Smooth motion, no relative friction moving parts. 4. High power-to-weight ratio and volume ratio, high energy-conversion efficiency. 5. It is not easy to generate heat or other harmful substances during operation. 6. They can realize multi-degree-of-freedom movement and operation. 7. Low price, convenient maintenance and wide application fields [2–4].

The PAMs are mainly composed of internal rubber and external diamond-like braided fibers, as shown in Figure 1. The outer woven mesh is composed of high-strength fibers. When the rubber tube is inflated, the rubber tube compresses the outer woven mesh due to elastic deformation. Due to the high rigidity of the woven mesh, it can only be radially deformed. The diameter becomes thicker and the length is shortened. Therefore, it is usually possible to adjust the contraction and elongation of the artificial muscle by changing the amount of air pressure in the rubber tube.

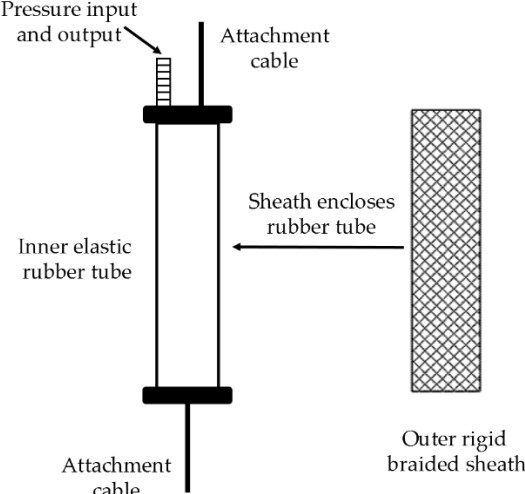

**Figure 1.** Structure of a pneumatic artificial muscle (PAM) [1].

Traditional rehabilitation machines usually use rigid electric motors as the drive, which easily cause uncomfortable rehabilitation. The ideal rehabilitation machine needs to be lightweight, highly safe and compliant, so the PAM is the most suitable for the drive of a rehabilitation machine. The purpose of this study is to develop a novel joint mechanism that combines a single-PAM with a torsion spring inside the joint mechanism and a rotary potentiometer to design a one degree-of-freedom (1-DOF) robotic lower limb system, which is designed to simulate the joint movement of human muscles and help the patient restore the function of the joint.

Because the pneumatic muscle actuator is a highly non-linear, high-order system, it is difficult to achieve precise control and affect the effectiveness of rehabilitation. Therefore, this paper proposes a fuzzy sliding mode controller (FSMC) for path-tracking control of a 1-DOF robotic lower limb system. Experiments show that this proposed control method can achieve the expected tracking performance and robustness.

Recently, conventional theoretical studies of robots that are controlled by PAMs, including industrial robots and wearable/rehabilitation robots, have shown similarities with the motion of the human limb. Sliding-mode control [5], adaptive self-organizing fuzzy sliding mode control [6], Proportional-integral-derivative (PID) control [7] and active force with fuzzy logic control [8] have been used to control a two-joint planar limb with four pairs of opposing PAMs. The simulation and experiment spatial-tracking responses of the end-effector, including sinusoidal spline, vertical line and circle motion, were achieved.

A PAM requires a precise actuator control. Adaptive control [2,9], learning vector quantization neural network control [10], variable structure control [11], energy-saving control [12], robust PID control [13] and self-organizing fuzzy control [14] have been used to control the classified system. The result is smooth actuator motion in response to step and/or sinusoidal inputs.

Kato et al. [15] developed a 4-DOF pneumatic robot limb that can be used as a gripper and incorporates two 2-DOF modules as the slave system. A main controller system that allows intuitive control was also developed. Robots that support daily life represent an evolution in robotics. Kobayashi et al. [2] developed a wearable robot for physical support or human assistance. This used a new connection mechanism and a smooth shoulder movement. Muramatsu et al. [16] developed a wearable muscle suit, that enabled manual workers to lift weights and gave quantitative evaluation results for direct and physical motion support.

This study proposes a 1-DOF robotic lower limb system that is driven by a novel single-PAM joint mechanism combined a single-PAM with a torsion spring. In the control system, model-free FSMC tracking is firstly used to design the PAM's path tracking controller, and finally implemented and verified experimentally for different paths. The experimental response results show that the innovative single-PAM with torsion spring pneumatic muscle mechanical arm system architecture can also achieve the same effect as the design of dual-PAM-driven joint antagonist muscles.

## 2. Mechanism Design of Novel Single-Pneumatic Artificial Muscle (PAM) System

The main purpose of this research is to develop an innovative pneumatic artificial muscle manipulator system, which will be applied to the lower limb portable assistive system in the future. In order to provide rehabilitation or for the patient to achieve home care or have a portable lower limb accessory rehabilitation system, it is necessary to design an assistive device that is easy to carry, light, small, of low-cost, and easy to operate. Unlike most previous PAM mechanism designs used in robotic arms, a pair of PAMs were used to drive a joint with a gear or a wheel. The design of the dual-PAMs antagonistic muscle mechanism results in a bulky overall structure, and the dual-PAMs also increase overall manufacturing costs. Therefore, the novel pneumatic artificial muscle manipulator system proposed by this study mainly uses a single-PAM with a torsion spring to drive a joint. The mechanism design of this single-PAM with a torsion spring replaces the mechanism design of using two PAMs with gears or wheels, and it is possible to achieve the same effect of driving the joint like dual-PAM mechanisms by combining a suitable control strategy.

In response to this mechanism, that is to be applied to portable lower-limb assists which usually have a multi-axis degree-of-freedom joint mechanism, the mechanical design of the single-PAM-driven joint reduces the overall auxiliary product and reduces costs. However, because the dual-PAM-driven joint has a better control response, in order to be provided with the same response as a dual-PAM-driven joint, a torsion spring embedded in the joint in parallel with a function of release energy and storage energy is developed to compensate for the lack of response of a single-PAM. Through experiments, the mechanism design of a single-PAM with a torsion spring can achieve a control response like the dual-PAMs, but the overall mechanism volume can also be reduced, be lighter and be of lower cost.

In order to drive a joint, this study proposes an innovative single-PAM with a torsion spring mechanism design, the stretching and contraction actuation of the PAM drives the torsion spring parallel to the joint to simulate the flexion and extension of the human joint antagonist muscle. By adjusting the internal pressure of the PAM so that its length changes according to the pressure, and drive the torsion spring to rotate the joint, the design concept diagram is shown in Figure 2.

As the PAM is inflated, the PAM length becomes shorter. In contrast, the PAM length is elongated as the PAM is deflated. However, since the PAM is only actuated in a single direction, the PAM length becomes longer when the air is deflated, but does not have a pulling force. Therefore, most of the PAM-driven joint designs in current literatures used dual-PAMs to achieve antagonism-driven function. In order to improve the inability of PAM to provide strength at the end when it is deflated, this study

proposes to replace the dual-PAM-driven mechanism design with a single PAM with a torsion spring embedded inside the joint that has the function of releasing energy and storing energy.

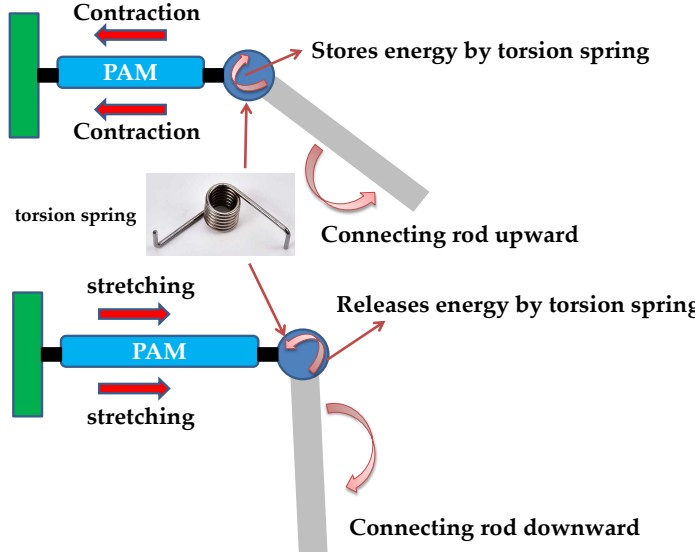

**Figure 2.** The mechanical design of the human robotic limb joint.

The torsion spring has the advantages of:

- Shock absorption: when the mechanical sudden impact load occurs, the spring can absorb the vibration energy to make the vibration more moderate.
- Force generation: the most common function of the spring applied to the machine is to use the elasticity of the spring to generate force or torque to maintain contact between the parts.
- Energy storage: the spring can generate elastic energy due to deformation, so the spring can be used to store energy.
- Force measurement: when the spring is deformed, the amount of deformation maintains a certain proportional relationship with the external force. With this characteristic, it can be used to measure the magnitude of the external force.

In this study, the spring mechanism is designed to use the torsion spring as the energy absorption of the PAM during elongation to generate the reaction force required for the joint rotation. The physical diagram and mechanism installation diagram are shown in Figure 3. The helical springs of the "Helical Torsion Spring (Product made by Spring Industries Co., Ltd, New Taipei, Taiwan)" are the same as the spiral compression springs or tension springs, however, the spring wire at the ends extends along the tangential direction of the spiral. When the spring is subjected to the torsional load, it will be deformed by the torque generated by the shaft of the spring.

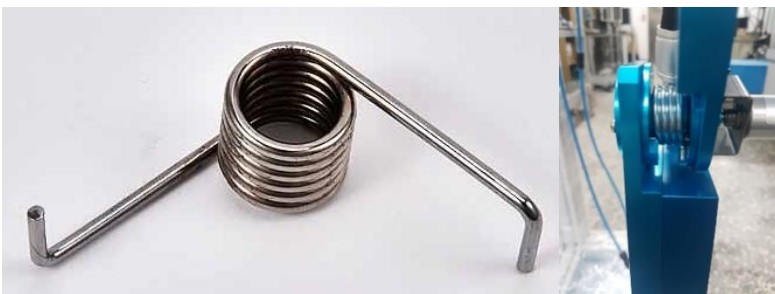

**Figure 3.** Torsion spring and installation mechanism.

From a biological structure point of view, human joints can come from the interaction of independent active tendons and antagonistic tendons. Therefore, the design of this study is based on the active and antagonistic role of human biomechanical muscle fibers in the life-like/antagonistic joint drive. A single-PAM with a torsion spring simulating linear muscle contraction is used to drive a single joint to achieve the joint position control.

The mechanical design concept of the bionic joint is shown in Figure 4. To simplify the biomechanical model, a single joint is driven by a single-PAM with a torsion spring. The joint angle is measured by a precision potentiometer and controlled by a designed controller to achieve angular control.

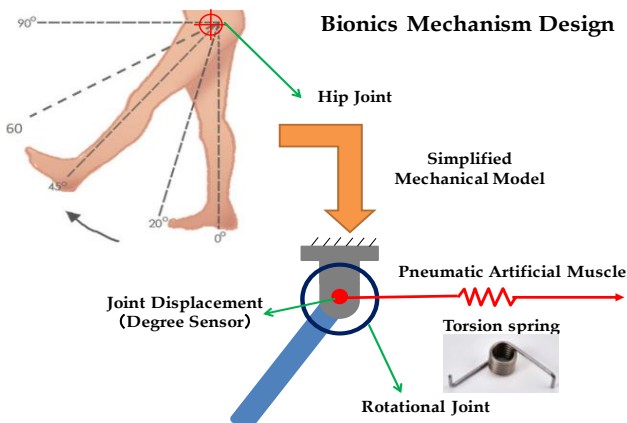

**Figure 4.** Design concept of the bionic joint driving by a single-PAM with a torsion spring.

## 3. Layout of Test Rig

The schematic model shows that the 1-DOF robotic lower limb system is driven by a single-PAM with torsion spring to achieve rotational motion, as shown in Figure 5. In this section, the 1-DOF robotic lower limb system test rig layouts are introduced. All the experiments are conducted using Real-Time Windows Target (RTWT) in the MATLAB environment. RTWT supports the interface card and automatically generates C code and executable files from SIMULINK models. The generated executable files run in real time on the Windows-based PC. Thus, the environment allows easy design and hardware-in-loop test of the control system.

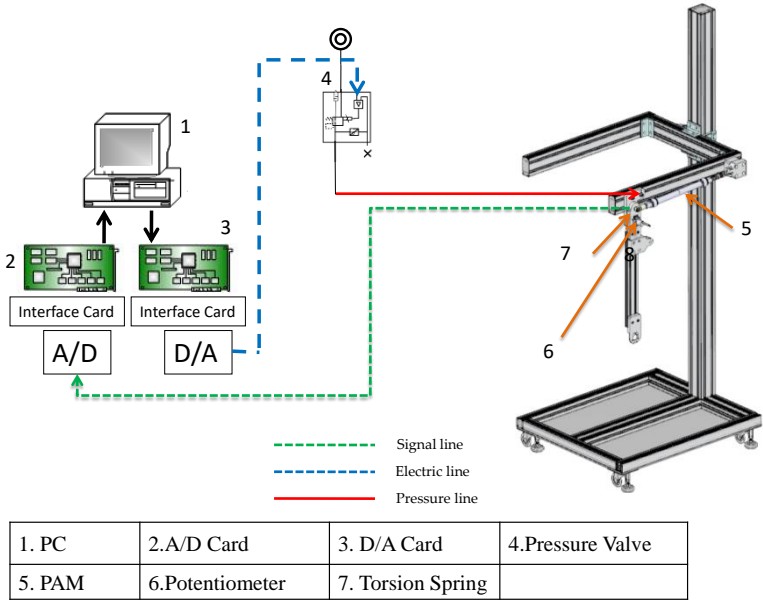

| 1. PC | 2.A/D Card | 3. D/A Card | 4.Pressure Valve |
|-------|------------|-------------|------------------|
| 5. PAM | 6.Potentiometer | 7. Torsion Spring | |

**Figure 5.** Test rig layout of the one degree-of-freedom (1-DOF) robotic lower limb system.

The main objective of this study is to develop a basis of our future research in the application of PAM on rehabilitation. The 1-DOF robotic lower limb system similar to human lower limb joint shown in Figure 5 has been set up. The mechanism design and the system layout are shown in Figure 6. Table 1 shows the specifications of the system.

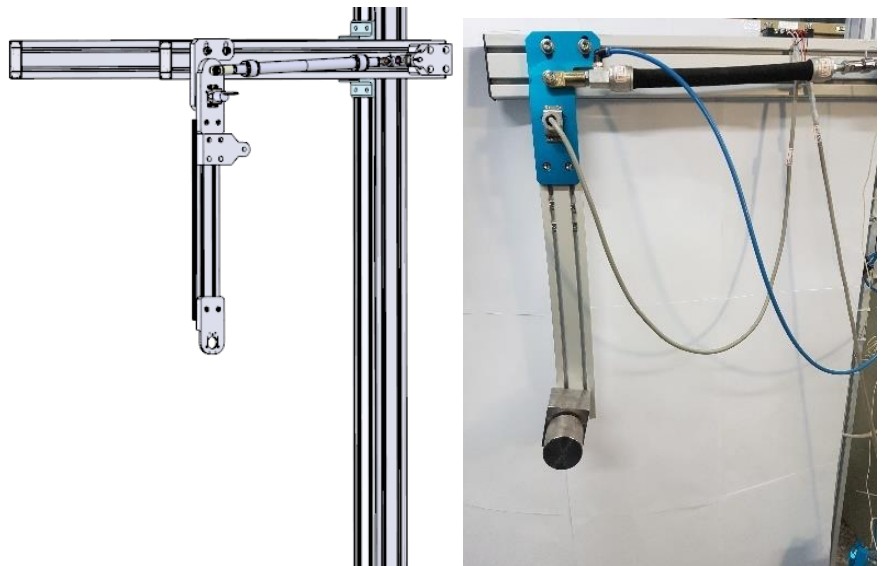

**Figure 6.** Photograph of the 1-DOF robotic lower limb system.

**Table 1.** Specification of the 1-DOF robotic lower limb system.

| Components | Company | Type | Specifications |
|---|---|---|---|
| A/D (Analog to Digital) Interface Card | Advantech | PCI-1720U | 4 channels12-bit D/A Output range: ±10 V |
| D/A (Digital to Analog) Interface Card | Advantech | PCI-1710 | 32 channels12-bit A/D Maximum input range: ±10 V |
| Pneumatic Muscle | FESTO | DMSP-20-200N-25cm | Max. stroke: 25% Max. force output: 1500 N Input pressure: 0~6 bar |
| Proportional Valve | FESTO | VPPM-8L-L-1-G14-0L6H-V1P | Input voltage: 0~10 V Output pressure: 0~6 bar |
| Potentiometer | Mitingen (Taiwan mitingen co., ltd.) | MTR22A | Range: 0~120 degree Output voltage: 0~5 V |

As shown in Figure 5, the 1-DOF robotic lower limb system is fixed to the bracket, and the joint of the lower limb is driven by a single-PAM with a torsion spring. The pressure in the PAM is controlled by the proportional pressure valve (FESTO, MPYE-5-1/8LF-010-B. Made by Festo AG & Co. KG Ruiter Straße 82 73734 Esslingen). A PC-based controller is used in a real-time environment with an analog to digital converter (Adventech, PCI-1720U, Taipei, Taiwan) and a digital to analog converter (Adventech, PCI-1710, Taipei, Taiwan). A torsion spring is used as the force and energy transmission. Table 2 shows the parameters of the torsion spring.

Because the angular motion range of the lower limb system hip joint is mainly in the cornal plane, this study only considers the extension and flexion movement of the hip joint. According to the literature [17], the hip joint extension range of motion angles is 0 to 20 degrees and the range of motion angles is 0 to 45 degrees. In the proposed mechanism design, the lower limbs of the robot can move from the vertical line by a maximum angle of about 90 degrees. The range of lower limb angles that can be designed by this body is in line with the requirements of lower limb assist devices.

**Table 2.** Parameters of the torsion spring.

| Parameters | Value | Unity |
|---|---|---|
| Wire diameter | 3.5 | mm |
| Center diameter | 26.5 | mm |
| Number of laps | 4 | loop |
| Installation angle | −10 | degree |
| Installation torque | 794 | N·mm |
| Free angle | 0 | degree |
| Working angle | 54 | degree |
| Working torque | 4280 | N·mm |

## 4. Controller Design

Since the PAM is a highly non-linear actuator, it is not easy to establish an accurate mathematical model. A FSMC that does not require a system model is used. In the experiment, the path tracking control of various positions of the lower limb joint with no-load is realized, and then an additional load added on the end of terminal of slider is used to position the trajectory, and the motion control of the 1-DOF robotic lower limb system under different trajectories is performed.

### 4.1. Control Strategy

In the 1-DOF robotic lower limb system, the controller is developed by the traditional PID control and the FSMC. The PID controller is used in this paper to compare the results with FSMC. The FSMC combines the traditional fuzzy theory and the sliding theory to reduce the input variables of the fuzzy controller by reducing the large fuzzy rule base of the traditional fuzzy rules, and combined with the sliding surface system can effectively suppress the system response of the non-linear term and improve the stability of system control.

In order to effectively simplify the fuzzy rule base without sacrificing control performance, this paper uses FSMC. The general fuzzy system judges the system response by two input variables including the error $e$ and the error rate $\dot{e}$, when establishing the fuzzy rule base. The fuzzy sliding surface system combines the input variable error $e$ and the error rate $\dot{e}$ into a sliding surface $\sigma = \alpha \cdot e + \dot{e}$, which replaces the error $e$ and the error rate $\dot{e}$ into a new input variable. Thus, the original two-dimensional fuzzy rule base is changed to the one-dimensional fuzzy sliding surface rule base, as shown in Figure 7.Where $X_{set}$ is the desired angle, $G_u$ is the scale factor, the control amount of the fuzzy controller after defuzzification can be given a proportional gain according to the system requirements to improve the actual response of the controller, U is the control output after defuzzification, s is a differentiator, and x is the actual output angle.

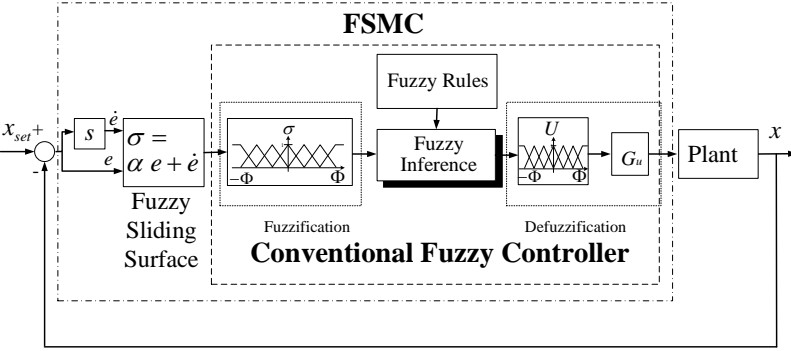

**Figure 7.** Block diagram of fuzzy sliding mode controller (FSMC).

### 4.2. Design of Fuzzy Sliding Mode Control (FSMC) for Path-Positioning Controller

The FSMC consists of two parts, a sliding surface estimator and a fuzzy logic controller [18–20]. Subtracting the target value from the feedback output signal produces the error $e$ and the error rate $\dot{e}$, multiplies the error term by the gain $\alpha$, and combines them into a sliding surface $\sigma$ as:

$$\sigma = \alpha \cdot e + \dot{e} \tag{1}$$

The traditional fuzzy controller adopts a two-dimensional fuzzy rule base whose control targets are $e \to 0$ and $\dot{e} \to 0$. After employing the sliding surface estimator, the two-dimensional input is replaced by the plane equation $\sigma = \alpha \cdot e + \dot{e}$, the error $e$ and the error rate $\dot{e}$ are reduced to an input value $\sigma$, and the control target becomes $\sigma = \alpha \cdot e + \dot{e} \to 0$, that is, the final output can also reach the set reference target.

Combining the above two points, we can know that in the fuzzy sliding mode control, the two-dimensional input of the fuzzy rule base can be judged as $e = 0$ and $\dot{e} = 0$ becomes $\sigma = \alpha \cdot e + \dot{e} = 0$, and the simplified fuzzy rule base will reduce its complexity and reduce the computer calculation in real time. As long as the sliding surface is maintained at $\sigma = \alpha \cdot e + \dot{e} = 0$, the system will eventually converge to the set ideal target value [21].

The value of $\alpha$ is the slope of the fuzzy sliding surface $\sigma$. The key of the fuzzy sliding surface planning is the selection of the value of $\alpha$. Therefore, the selection of the value of $\alpha$ will directly affect the convergence speed of the control system and the performance of the controller, and determine the stability of the system. The determination of the value of $\alpha$ must depend on the system characteristics of the control system.

The value of $\Phi$ is the convergence boundary of the fuzzy sliding surface $\sigma = \alpha \cdot e + \dot{e} = 0$, as shown in Figure 8. The selection of the value of $\Phi$ is determined according to the control input of the fuzzy sliding surface $\sigma = \alpha \cdot e + \dot{e}$. Therefore, the value of $\Phi$ must be selected suitably according to the control input and the actual system operation.

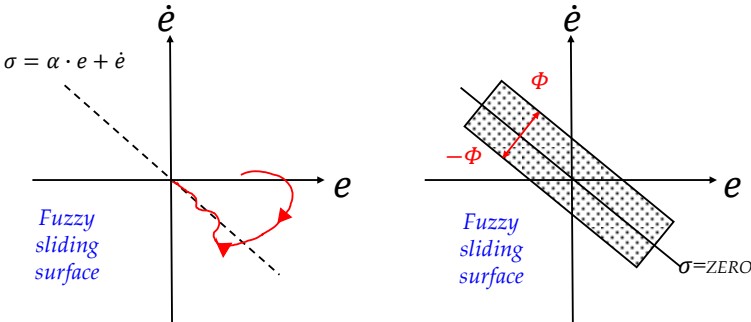

**Figure 8.** Fuzzy sliding surface.

The control parameters such as $\alpha$, $\Phi$, $u_0$, $G_u$ have certain influence on the dynamic response of the control system. Therefore, the selection of parameters must be adjusted according to the actual needs and characteristics of the system.

The establishment of the membership function first sets the membership function of the input variable $\sigma$ is $T(\sigma) = \{NB, NM, NS, ZR, PS, PM, PB\}$, and the membership function of the output variable $u$ is $T(u) = \{NB, NM, NS, ZR, PS, PM, PB\}$, according to the established fuzzy set combined with the sliding mode control theory, the relationship between the fuzzy sliding surface $\sigma$ and the control input $u$ can be obtained.

According to the output state of the system, we can set the following fuzzy rules shown in Table 3. NB, NM, NS, ZR, PS, PM, and PB are all labels of the fuzzy set, and the membership function of the input variable $\sigma$ of the fuzzy sliding surface is shown in Figure 9, the membership function of the output variable $u$ of the fuzzy sliding surface is shown in Figure 10.

**Table 3.** Fuzzy rules.

| Rules | $\sigma$ | $u$ |
|-------|----------|-----|
| R$_1$ | If $\sigma$ is PB | Then $u$ is PB |
| R$_2$ | If $\sigma$ is PM | Then $u$ is PM |
| R$_3$ | If $\sigma$ is PS | Then $u$ is PS |
| R$_4$ | If $\sigma$ is ZR | Then $u$ is ZR |
| R$_5$ | If $\sigma$ is NS | Then $u$ is NS |
| R$_6$ | If $\sigma$ is NM | Then $u$ is NM |
| R$_7$ | If $\sigma$ is NB | Then $u$ is NB |

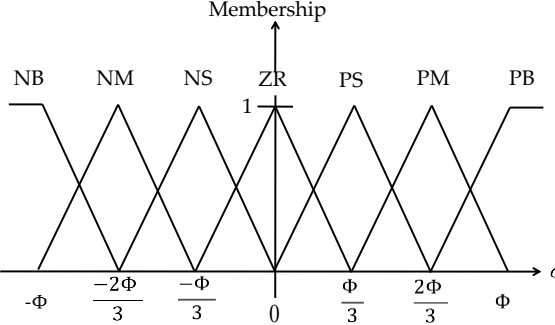

**Figure 9.** The membership function of the input variable $\sigma$ of the fuzzy sliding surface.

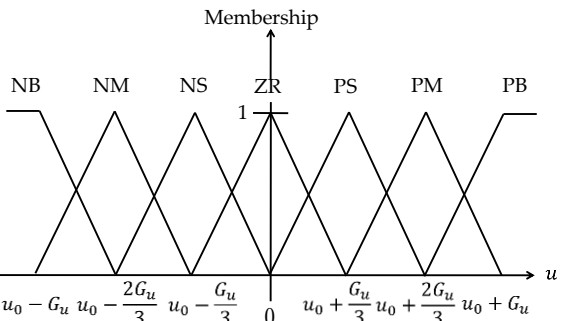

**Figure 10.** The membership function of the output variable $u$ of the fuzzy sliding surface.

Therefore, the relationship between the fuzzy sliding surface $\sigma$ and the control input $u_f$ is expressed by the following Equation (2).

$$u_f = u_0 - u_d$$
$$u_d = G_u(x,t){\cdot}sig(w) \tag{2}$$
$$w = \frac{\sigma}{\Phi}$$

where $u_0$ can be regarded as the offset of the control signal input, and sigmoid function $sig(w)$ can be described as:

$$sig(w) = \begin{cases} -1 & if\ z < -1 \\ \dfrac{-\left(21w^2+27w+10\right)}{2(9w^2+12w+5)} & if\ -1 \leq z < -\frac{2}{3} \\ \dfrac{-\left(15w^2+9w+2\right)}{2(9w^2+6w+2)} & if\ -\frac{2}{3} \leq z < -\frac{1}{3} \\ \dfrac{-\left(9w^2-w\right)}{2(9w^2+1)} & if\ -\frac{1}{3} \leq z < 0 \\ \dfrac{\left(9w^2-w\right)}{2(9w^2+1)} & if\ 0 \leq z < \frac{1}{3} \\ \dfrac{\left(15w^2-9w+2\right)}{2(9w^2-6w+2)} & if\ \frac{1}{3} \leq z < \frac{2}{3} \\ \dfrac{\left(21w^2-27w+10\right)}{2(9w^2-12w+15)} & if\ \frac{2}{3} \leq z < 1 \\ 1 & if\ z \geq 1 \end{cases} \tag{3}$$

This study uses the Mamdani method as the inference principle of the system. Since the output of the controller requires a crisp value, it is necessary to defuzzify the fuzzy set obtained by the inference. In the method of defuzzification, the centroid method is used as the defuzzification rule. Table 4 shows the rule base of the fuzzy sliding surface, where the fuzzy sliding surface requires only seven rules, thus greatly simplifying the traditional rule base. Therefore, the use of fuzzy sliding mode controller can greatly simplify the design difficulty of the fuzzy rule base. Figure 11 shows the control block diagram for the 1-DOF robotic lower limb system driven by the novel single-PMA with a torsion spring. The parameters of the FSMC designed in this study are shown in Table 5.

**Table 4.** Traditional fuzzy sliding rule base.

| | $\sigma = \alpha e + \dot{e}$ | | | | | | |
|---|---|---|---|---|---|---|---|
| | **NB** | **NM** | **NS** | **ZR** | **PS** | **PM** | **PB** |
| $u$ | NB | NM | NS | ZR | PS | PM | PB |

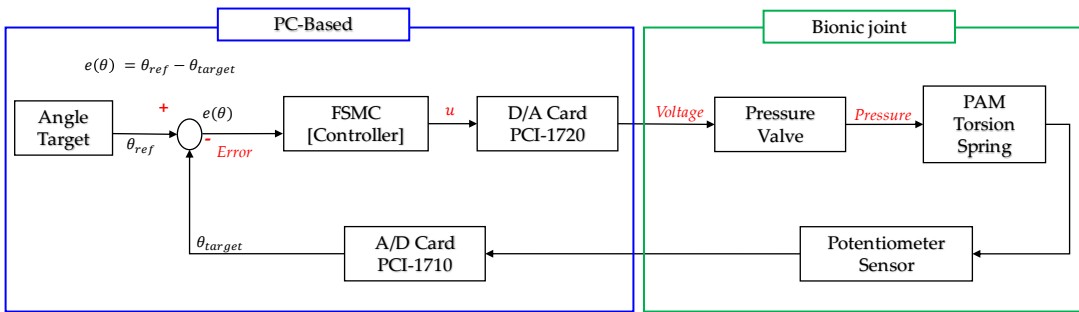

**Figure 11.** The control block diagram for the 1-DOF robotic lower limb system driven by a single-PMA with a torsion spring.

**Table 5.** Control parameters of FSMC for path tracking experiments.

| Parameters | Description | Value |
|---|---|---|
| $\alpha$ | The slope of the fuzzy sliding surface | 1 |
| $\sigma$ | The fuzzy sliding surface | $\sigma = 1 \cdot e + \dot{e}$ |
| $u_0$ | The offset of the control signal input | 0 |
| $\Phi$ | The convergence boundary of the fuzzy sliding surface | $\Phi = 1/30000$ |
| $T(\sigma)$ | The membership function of the input variable $\sigma$ | $T(\sigma) = \{NB, NM, NS, ZR, PS, PM, PB\}$ $= \{-0.77, -0.52, -0.256, 0, 0.256, 0.52, 0.77\}$ |
| $T(u)$ | The membership function of the output variable $u$ | $T(u) = \{NB, NM, NS, ZR, PS, PM, PB\}$ $= \{0.115, 0.244, 0.372, 0.5, 0.628, 0.756, 0.885\}$ |
| $G_u$ | The scale factor | 5.8 |

## 5. Experiments

In this section, the path positioning and tracking experiments of the 1-DOF robotic lower limb system driven by a single-PAM with a torsion spring with the FSMC are implemented. The experimental results of the cases without additional loading mass and with a 2 kg loading mass added on the 500 mm length slider compared with traditional PID controller are performed with the sampling time of 0.001 s.

### 5.1. Step Response Experiment

The step response for the experiment without additional loading is shown in Figure 12. The results with a 2 kg loading mass added on the 500 mm length slider are shown in Figure 13.

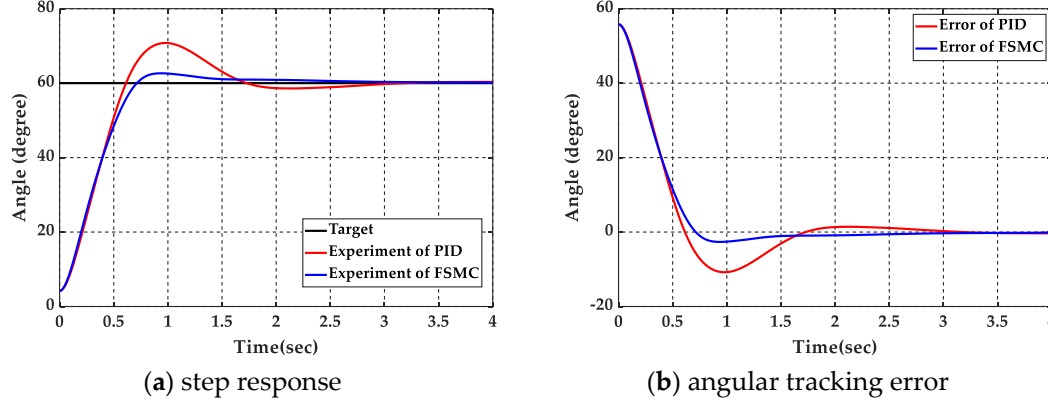

**(a)** step response

**(b)** angular tracking error

**Figure 12.** Comparison of experimental results for step input for the 1-DOF robotic lower limb system driven by a single-PAM with a torsion spring without loading between PID (Proportional–Integral–Derivative) controller and FSMC: (**a**) the step response; (**b**) the angular tracking error.

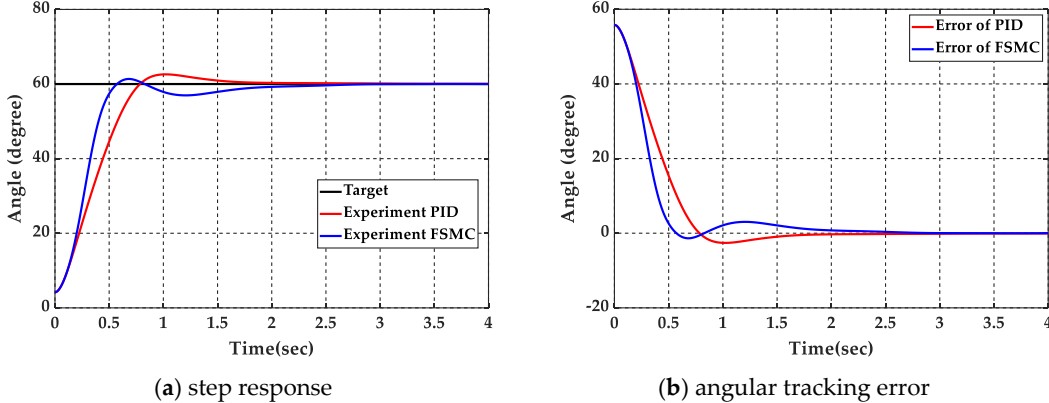

**(a)** step response

**(b)** angular tracking error

**Figure 13.** Comparison of experimental results for step input for the 1-DOF robotic lower limb system driven by a single-PAM with a torsion spring with a 2 kg loading mass between PID controller and FSMC: (**a**) the step response; (**b**) the angular tracking error.

The experimental results show that the steady angular tracking error without additional loading is about 0.085 degrees, and with a 2 kg loading mass is 0.0338 degrees by using the PID controller, respectively. By FSMC the steady angular tracking error can achieve 0.003 degrees without additional loading and 0.00338 degrees with a 2 kg loading mass, respectively. The performance of the step response, including transient response and steady state response, is summarized in Table 6.

**Table 6.** Performance of step response.

| Parameters / Controller | Loading | Rising Time (s) | Settling Time (s) | Peak Time (s) | Overshoot (%) | Steady State Error (Degree) |
|---|---|---|---|---|---|---|
| PID Controller | No | 0.4313 | 2.4066 | 0.9780 | 18.1499 | 0.0850 |
| FSMC | | 0.4873 | 1.4733 | 0.9730 | 4.4194 | 0.0030 |
| PID Controller | Yes | 0.5370 | 1.4254 | 1.0210 | 4.2278 | 0.0566 |
| FSMC | | 0.3389 | 1.7927 | 0.6820 | 2.2125 | 0.0338 |

### 5.2. Triangular Wave Path-Tracking Control Experiment

The experiments of the path tracking control are compared with using traditional PID controller and FSMC without additional loading and with a 2 kg loading mass added on the 500 mm length slider, respectively. The path-tracking control is set as the triangular wave path with different periods and the amplitude is set to ±10 degrees.

### 5.2.1. Without Additional Loading

The triangular wave path-tracking control response with periods of 20 s and 10 s for the experiment without additional loading by using the PID controller and FSMC are shown in Figures 14–17, including the angular path response, the angular tracking error and control signal.

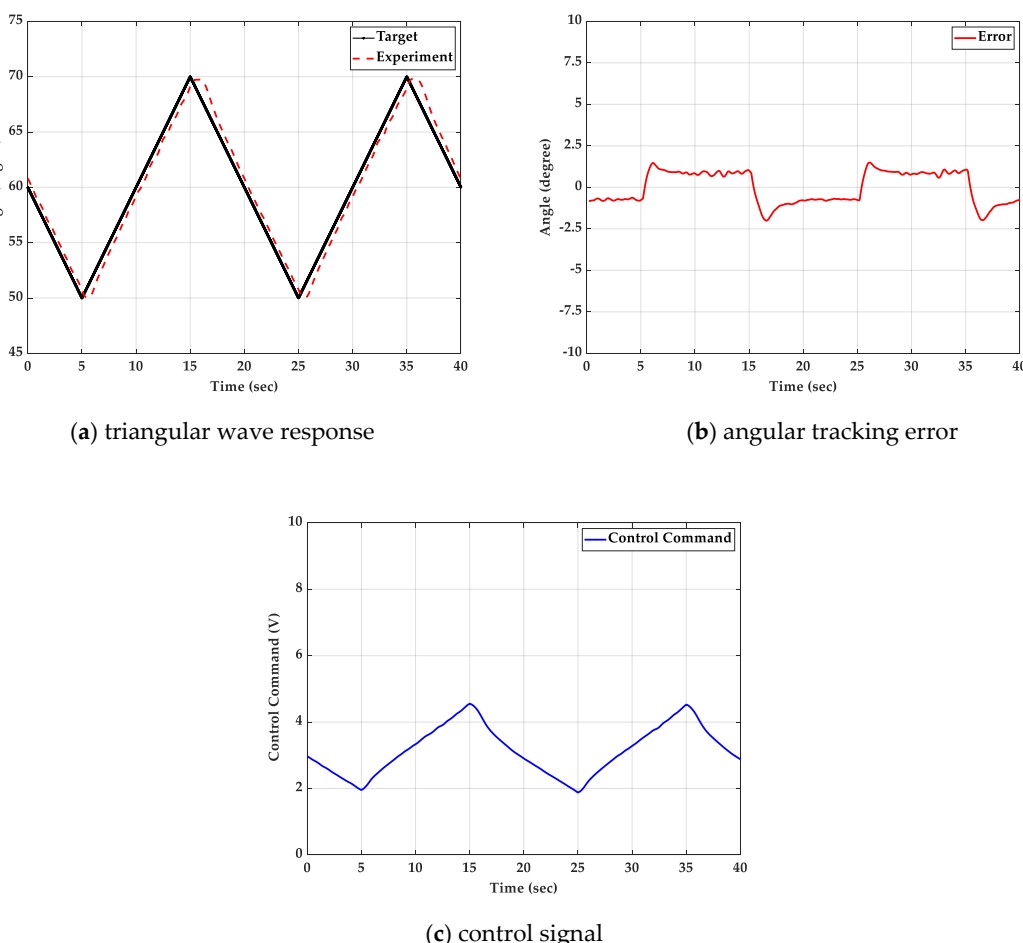

(**a**) triangular wave response

(**b**) angular tracking error

(**c**) control signal

**Figure 14.** Experimental results for triangular wave path tracking control with a period of 20 s for the 1-DOF robotic lower limb system driven by a single-PAM with a torsion spring without loading by using the PID controller: (**a**) step response; (**b**) angular tracking error; (**c**) control signal.

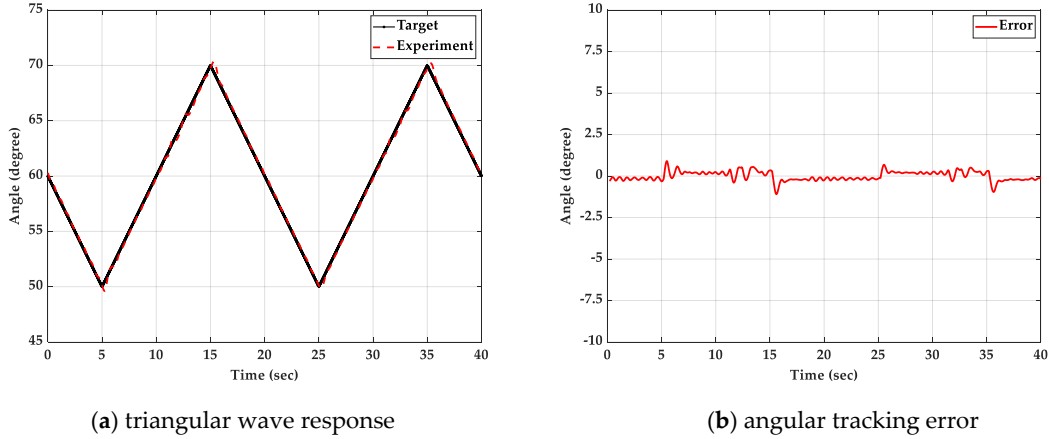

(**a**) triangular wave response

(**b**) angular tracking error

**Figure 15.** *Cont.*

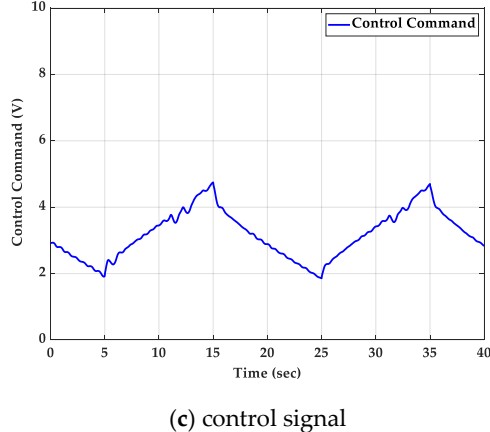

(**c**) control signal

**Figure 15.** Experimental results for triangular wave path tracking control with a period of 20 s for the 1-DOF robotic lower limb system driven by a single-PAM with a torsion spring without loading by using the FSMC: (**a**) step response; (**b**) angular tracking error; (**c**) control signal.

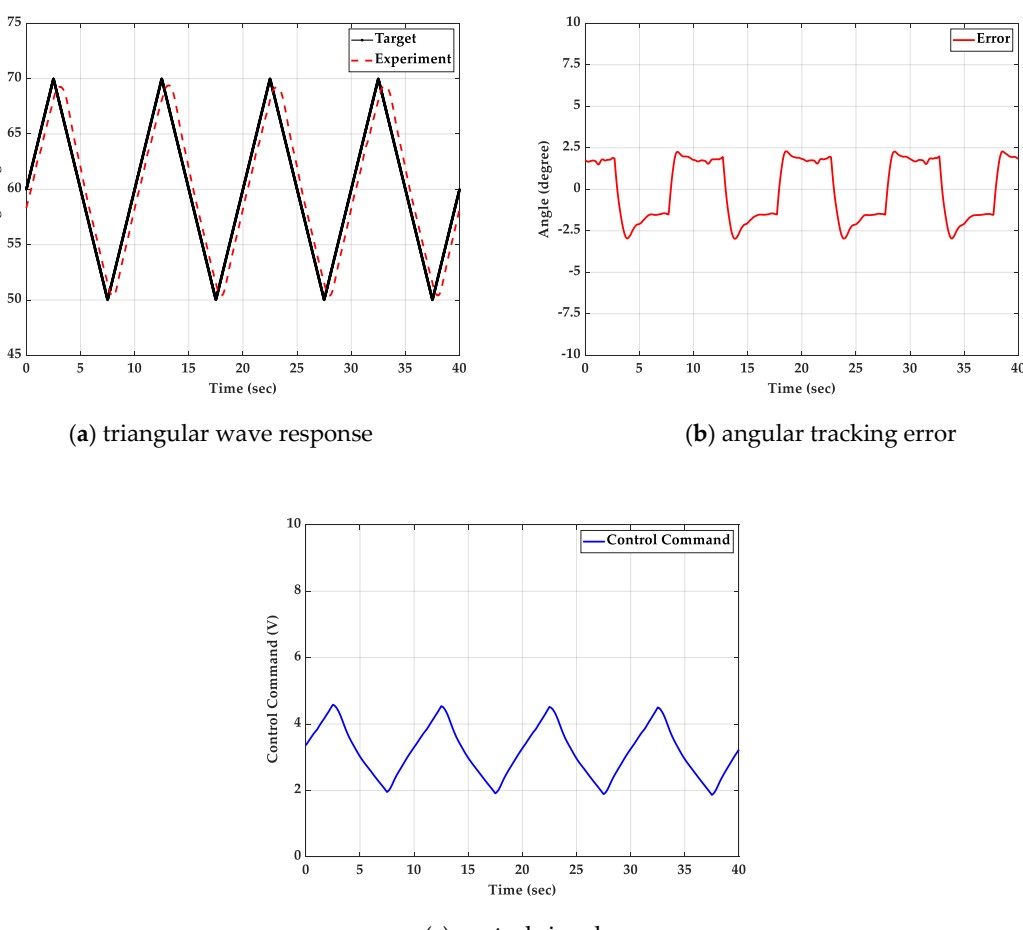

(**a**) triangular wave response          (**b**) angular tracking error

(**c**) control signal

**Figure 16.** Experimental results for triangular wave path tracking control with a period of 10 s for the 1-DOF robotic lower limb system driven by a single-PAM with a torsion spring without loading by using the PID controller: (**a**) step response; (**b**) angular tracking error; (**c**) control signal.

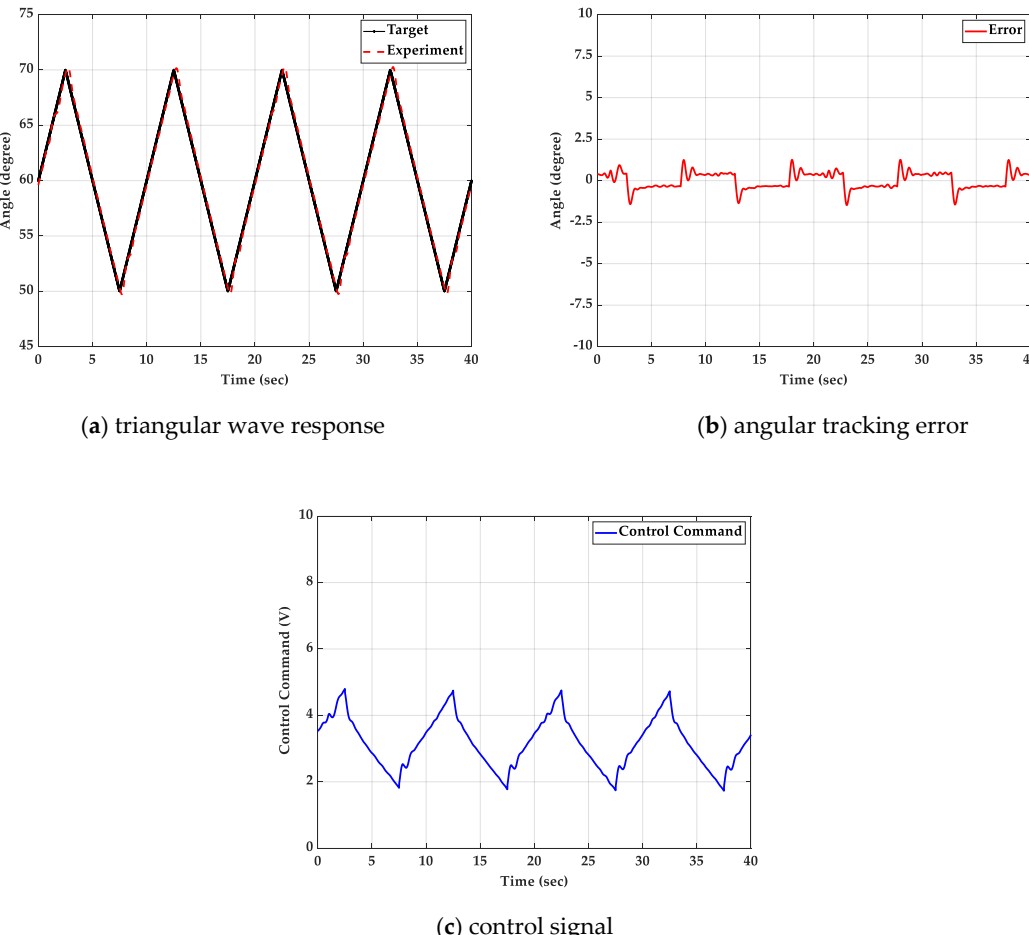

(**a**) triangular wave response　　　　　　　　　　(**b**) angular tracking error

(**c**) control signal

**Figure 17.** Experimental results for triangular wave path tracking control with a period of 10 s for the 1-DOF robotic lower limb system driven by a single-PAM with a torsion spring without loading by using the FSMC: (**a**) step response; (**b**) angular tracking error; (**c**) control signal.

In order to analyze and compare the results in detail, the integral of absolute error (IAE) is used to illustrate the accumulative error of the PID controller and FSMC under various experimental conditions. In the case of no load, the PID controller has an IAE of 64.1752 in a period of 20 s and an IAE of 105.278 in a period of 10 s. However, the FSMC can achieve an IAE of 30.4597 in period of 20 s and an IAE of 40.5492 in period of 10 s. Thus, FSMC can perform much better than PID control in this case.

### 5.2.2. With a 2 kg Loading Mass

The triangular wave path-tracking control response with periods of 20 s and 10 s for the experiment with a 2 kg loading mass added on the 500 mm length slide by using the PID controller and FSMC is shown in Figures 18–21. According to the experimental results in the case with a 2 kg loading mass, the PID controller has an IAE of 60.8661 in a period of 20 s and an IAE of 95.0987 in a period of 10 s. The FSMC can achieve an IAE of 38.2104 in a period of 20 s and an IAE of 55.435 in a period of 10 s. Therefore, FSMC has obviously better performance than the PID control.

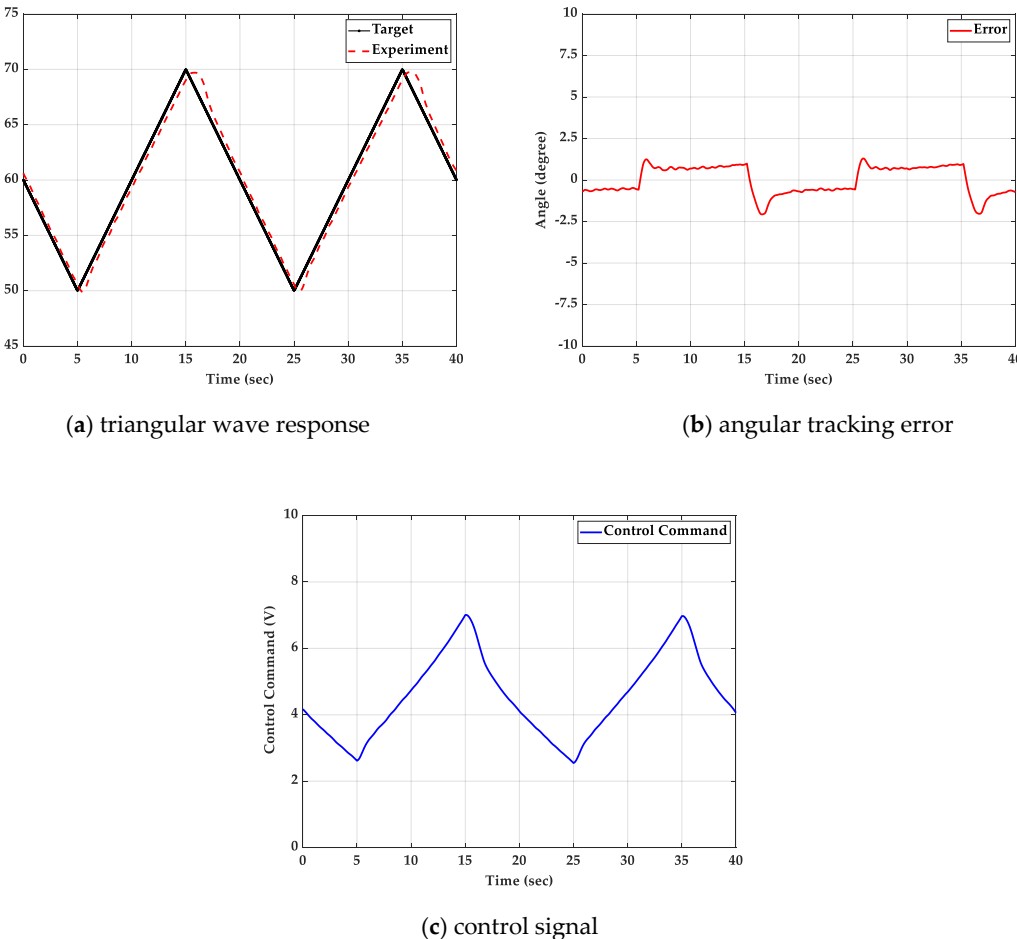

(**a**) triangular wave response

(**b**) angular tracking error

(**c**) control signal

**Figure 18.** Experimental results for triangular wave path tracking control with a period of 20 s for the 1-DOF robotic lower limb system driven by a single-PAM with a torsion spring with a 2 kg loading mass by using the PID controller: (**a**) step response; (**b**) angular tracking error; (**c**) control signal.

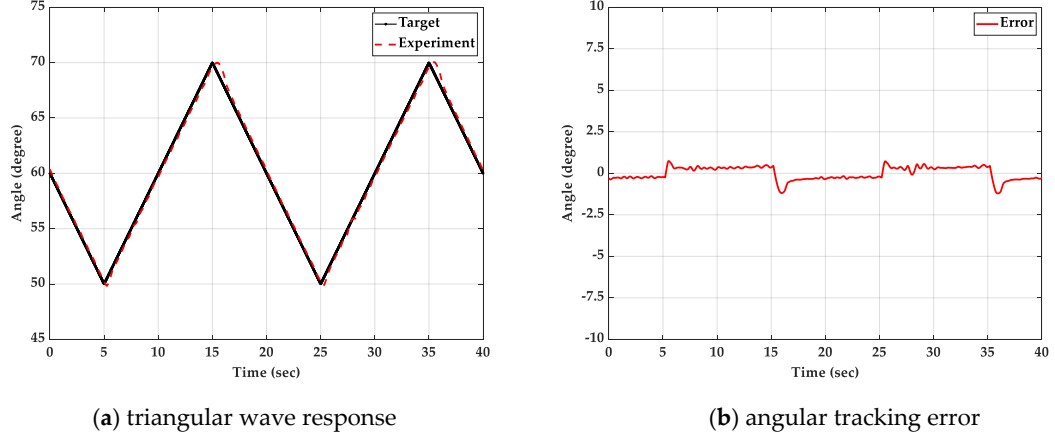

(**a**) triangular wave response

(**b**) angular tracking error

**Figure 19.** *Cont.*

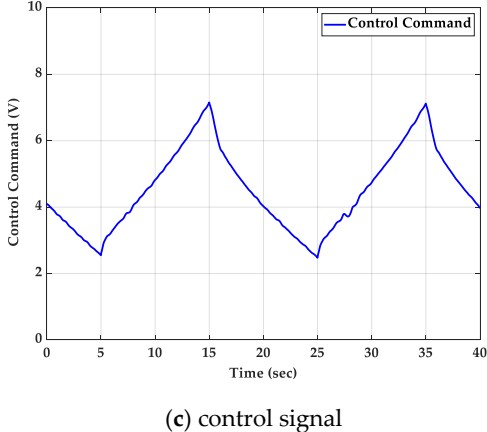

(**c**) control signal

**Figure 19.** Experimental results for triangular wave path tracking control with a period of 20 s for the 1-DOF robotic lower limb system driven by a single-PAM with a torsion spring with a 2 kg loading mass by using the FSMC: (**a**) step response; (**b**) angular tracking error; (**c**) control signal.

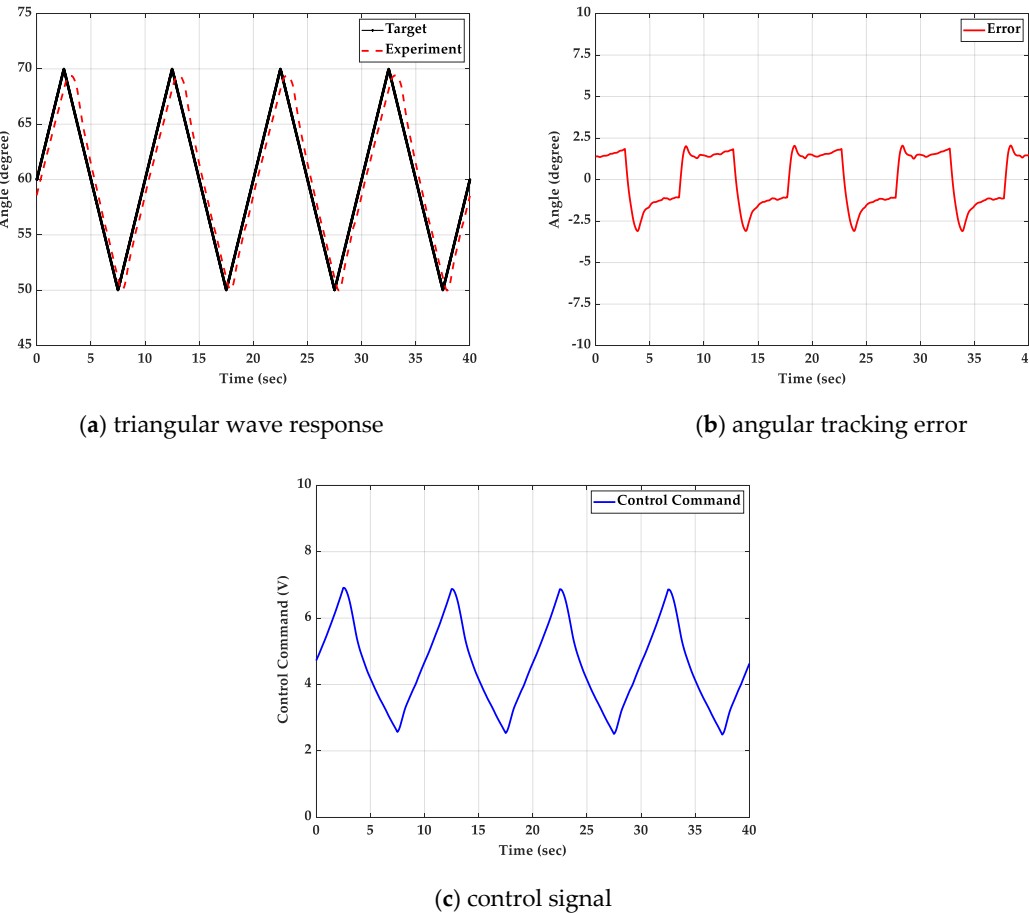

(**a**) triangular wave response                                    (**b**) angular tracking error

(**c**) control signal

**Figure 20.** Experimental results for triangular wave path tracking control with a period of 10 s for the 1-DOF robotic lower limb system driven by a single-PAM with a torsion spring with a 2 kg loading mass by using the PID controller: (**a**) step response; (**b**) angular tracking error; (**c**) control signal.

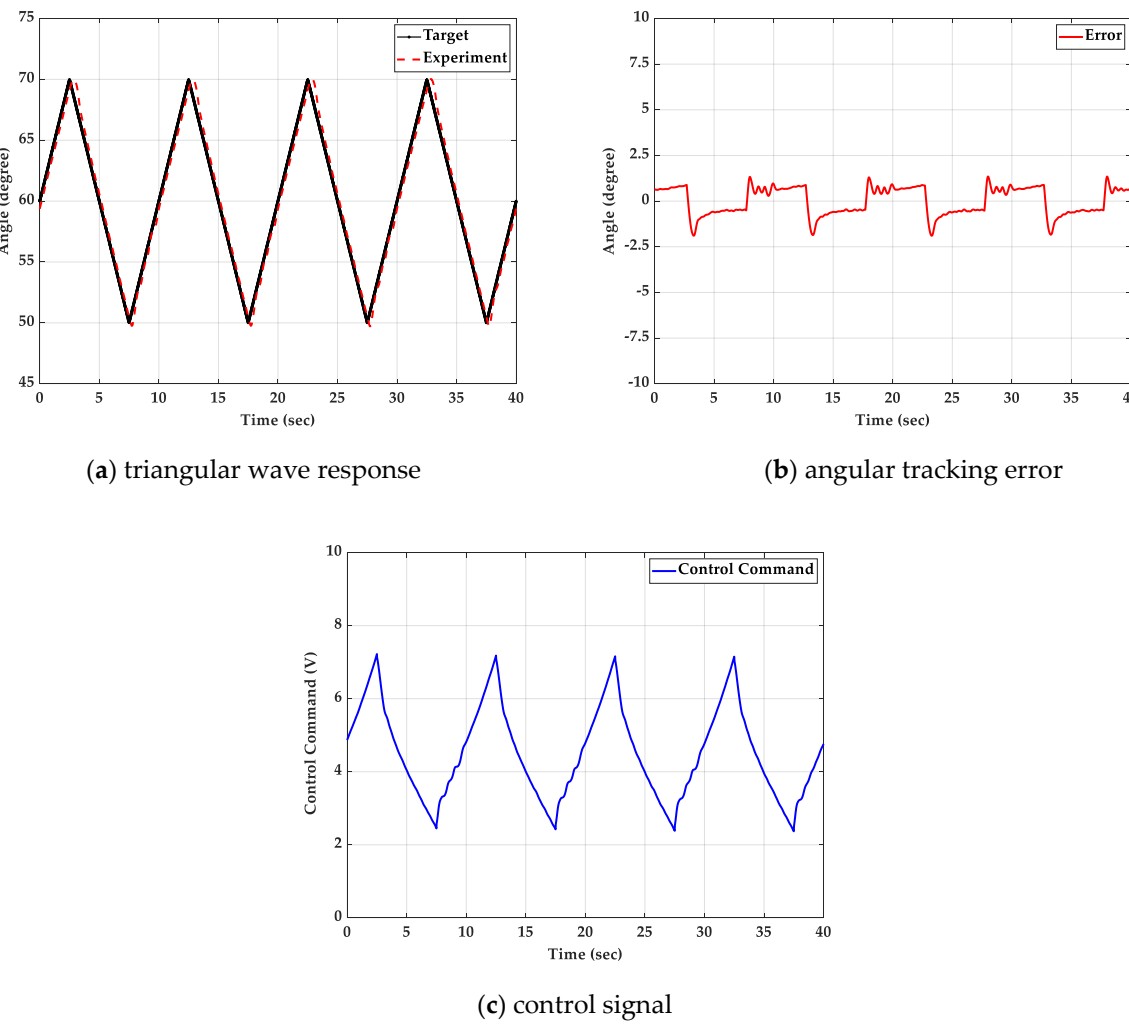

(**a**) triangular wave response

(**b**) angular tracking error

(**c**) control signal

**Figure 21.** Experimental results for triangular wave path tracking control with a period of 10 s for the 1-DOF robotic lower limb system driven by a single-PAM with a torsion spring with a 2 kg loading mass by using FSMC: (**a**) step response; (**b**) angular tracking error; (**c**) control signal.

### 5.3. Sinusoidal Wave Path-Tracking Experiment with a 2 kg Loading Mass

The experiment of the sinusoidal path-tracking control are compared with a traditional PID controller and FSMC with a 2 kg loading mass added on the 500 mm length slider. The path-tracking control is set as the sinusoidal wave path with different periods and the amplitude is set to ±10 degrees.

The sinusoidal wave path-tracking control response with frequency of 0.05 Hz and 0.1 Hz for the experiment with a 2 kg loading mass by using the PID controller and FSMC is shown in Figures 22–25, respectively.

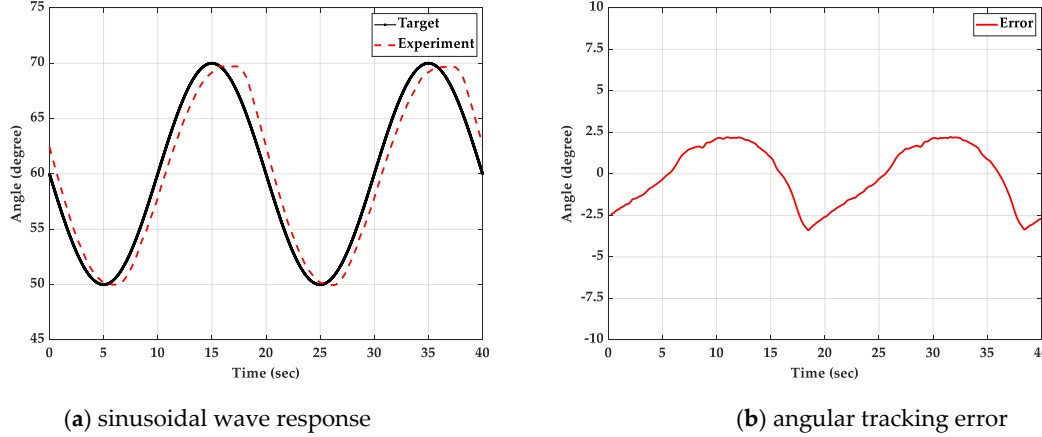

(**a**) sinusoidal wave response          (**b**) angular tracking error

**Figure 22.** Experimental results for sinusoidal wave path tracking control with frequency in 0.05 Hz for the 1-DOF robotic lower limb system driven by a single-PAM with a torsion spring with a 2 kg loading mass by using the PID controller: (**a**) the angular path response; (**b**) the angular tracking error.

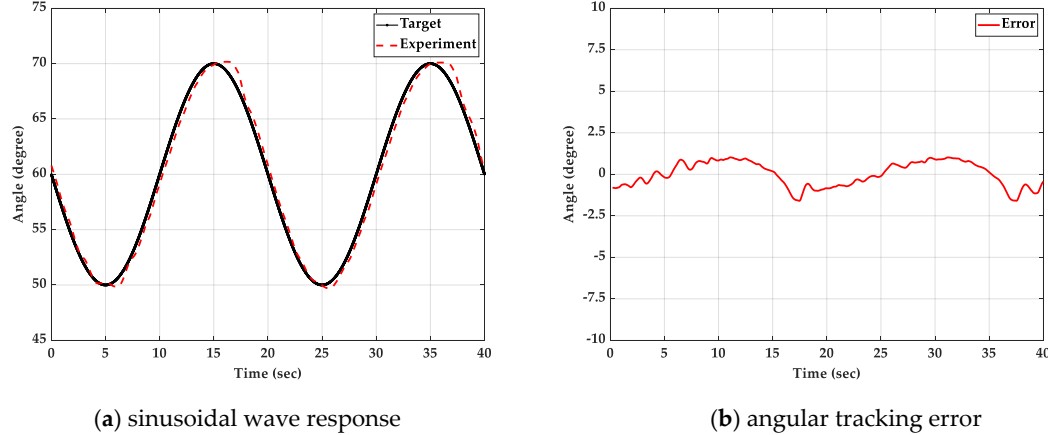

(**a**) sinusoidal wave response          (**b**) angular tracking error

**Figure 23.** Experimental results for sinusoidal wave path tracking control with frequency in 0.05 Hz for the 1-DOF robotic lower limb system driven by a single-PAM with a torsion spring with a 2 kg loading mass by using the FSMC: (**a**) the angular path response; (**b**) the angular tracking error.

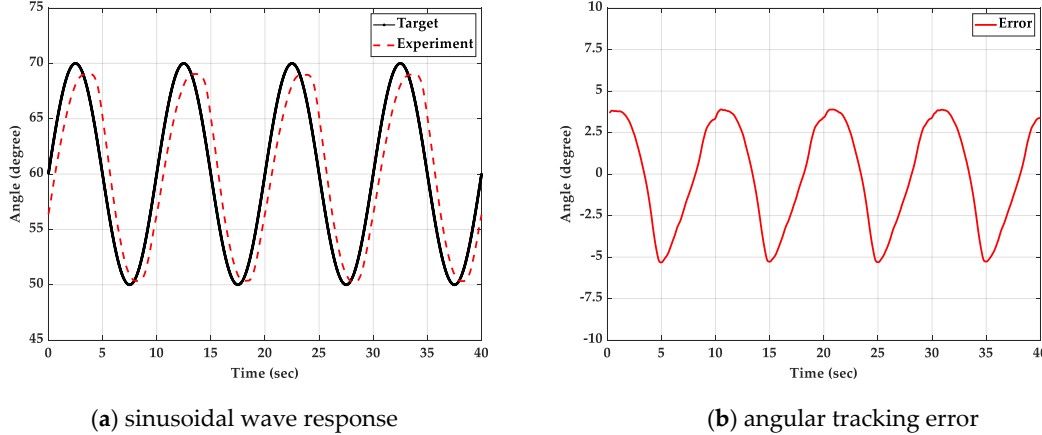

(**a**) sinusoidal wave response          (**b**) angular tracking error

**Figure 24.** Experimental results for sinusoidal wave path tracking control with frequency in 0.1 Hz for the 1-DOF robotic lower limb system driven by a single-PAM with a torsion spring with a 2 kg loading mass by using the PID controller: (**a**) step response; (**b**) angular tracking error.

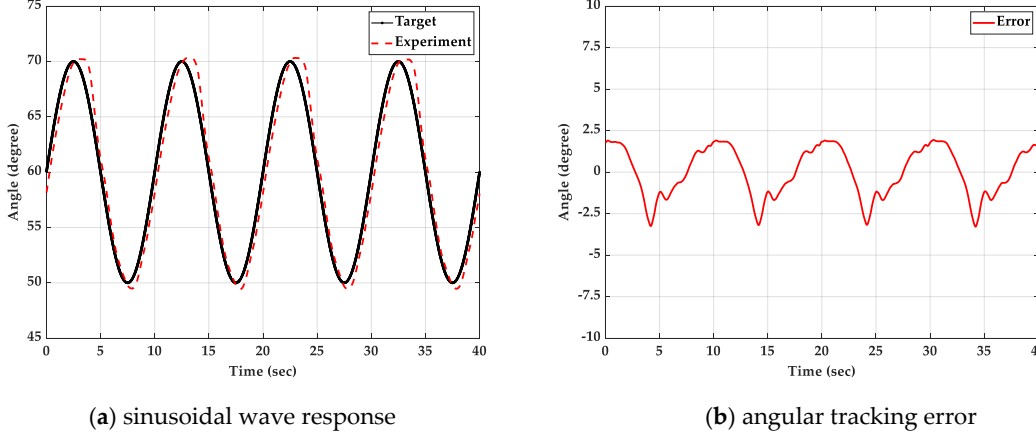

(**a**) sinusoidal wave response　　　　　　　　　　　　　　　(**b**) angular tracking error

**Figure 25.** Experimental results for sinusoidal wave path tracking control with frequency in 0.1 Hz for the 1-DOF robotic lower limb system driven by a single-PAM with a torsion spring with a 2 kg loading mass by using the FSMC: (**a**) step response; (**b**) angular tracking error.

According to the experimental results under the load of 2 kg loading mass, the PID controller has an IAE of 100.8331 in a frequency of 0.05 Hz and an IAE of 155.6086 in a frequency of 0.1 Hz. However, the FSMC can reach an IAE of 65.528 in a frequency of 0.05 Hz and an IAE of 93.5461 in frequency of 0.1 Hz. Moreover, the tracking error in both cases is within the range of 0.3 degrees by using FSMC, better than PID control.

Due to the compressibility of the air pressure and the non-linear elasticity of the rubber, the enthalpy changes are caused. Therefore, it is not easy to achieve good tracking performance for the 1-DOF robotic lower limb system. The traditional PID controller and the FSMC are adopted to overcome the nonlinear behavior of PAMs, and to compare and analyze the performance of the controller.

The traditional PID controller is applied to a robotic lower-limb system composed of a single-PAM. Compared with the FSMC, the control performance is poor, and the required stabilization time is longer in the step response, and the maximum overshoot is also large. In the sinusoidal wave response, the tracking error are far worse than the FSMC. The FSMC uses the sliding surface to simplify the two-dimensional fuzzy rule base into one-dimensional, which can not only shorten the calculation time, but also can display better robustness against changes in the external environment.

According to the above various experiments, the proposed novel single-PAM with a torsion spring-driven lower-limb accessory joint mechanism can be verified with satisfactory performance.

## 6. Conclusions

In this study, the 1-DOF robotic lower-limb system driven by the single-PAM with a torsion spring installed on the mechanism of joint by using a proportional pressure valve and controlled by model-free FSMC was implemented. In the experiment, the PID controller and the FSMC were used to control the single-PAM with a torsion spring to drive the robotic lower-limb system. This can test whether the control system can effectively achieve the position tracking control under different targets with different periods and waveforms. The FSMC adopts the sliding surface to simplify the fuzzy rule base, shorten the calculation time, and display better robustness against changes in the external environment, and thus with much better performance than PID control.

The lower-limb assist system is mainly used to help patients complete self-rehabilitation. In the overall control response of this study, the tracking angle error can be controlled within 0.5 degrees, and is sufficient for the accessory system. Through various experimental results, the novel single-PAM with a torsion spring applied to drive a single joint can achieve satisfactory responses in various positions and tracking control. Instead of dual-PMAs, the novel single-PAM with a torsion spring proposed by this study can reduce the size and complexity of the overall mechanism, reduce the production cost, and also improve the portability and convenience required for the wearable accessory.

**Author Contributions:** M.-H.C. conceived and designed the experiments. T.-C.T. performed the experiments and analyzed the data. T.-C.T. and M.-H.C. wrote the paper. All authors have read and agreed to the published version of the manuscript.

**Funding:** The research was sponsored in part by the Ministry of Science and Technology, Taiwan under the grant MOST 103-2221-E-002-160-MY2 and MOST 104-3113-E-002-016-CC2.

**Acknowledgments:** The authors would like to thank the Ministry of Science and Technology, Taiwan for the sponsor in part of this research.

**Conflicts of Interest:** The authors declare no conflict of interest.

## Abbreviations

The following abbreviations are used in this manuscript:

PAM    Pneumatic artificial muscle
FSMC   Fuzzy sliding mode control

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
