# Peer review of "Design and Control of a 1-DOF Robotic Lower-Limb System Driven by Novel Single Pneumatic Artificial Muscle"

_applsci, doi:10.3390/app10010043_

Round 1

Reviewer 1 Report

This paper is timely due to the current high level of interest in soft robotics and soft actuation systems. The paper is generally well written but needs a thorough proof reading as there are some occasional typos e.g. Ln 164 “session” should read “section”.

The paper describes an actuation method and controller for a single joint for a lower limb exoskeleton system. However, an exoskeleton is not presented and the technical specification of the system do not appear to be derived from the requirements of an exoskeleton. The system and controller are general purpose and could be applied to a range of robotic applications. I suggest the paper could be improved by removing reference to an exoskeleton and making it a more general paper on soft actuation and PAMs.

In the abstract and the main body of the paper there is repeated mention of the fact that “all PAM-driven joint designs in all current literature used dual-PAMs” (Ln 125). This is not a true statement, there are many examples when PAM have been used with a spring as the antagonist, Schulte’s paper from 1961 presents a joint using a single PAM and there have been numerous examples since e.g. in robot hands that use PAM. Claims that the work which presents a single PAM joint mechanism is novel is overstated and not substantiated by literature. Single muscles actuation is less common that having two PAM but there are many examples in literature.

Using a spring as an antagonist is less efficient than using two PAM as when a spring is use the muscle will always have to overcome the force of the spring during flexion. Where two muscle are used the extensor can relax during flexion. The paper should comment on this.

Also when using a spring antagonist it is not possible to vary the stiffness of the joint, which is possible when using two PAM. This paper should also comment on this.

The advantages of the torsion spring (Ln 132) all apply to the situation where a second PAM is used. A PAM can also absorb shock, generate force and store energy. This justification for using a spring instead of a PAM is weak and needs to be strengthened – maybe consider financial cost and the additional hardware needed to control a second PAM.

Table 1 and 2 do not seem like a good use of space, most of the information is already in the text so I recommend removing these tables.

In the experimental results plots there is large amounts of wasted space with the important areas of the graph being very small. For example in fig4(a) after 4secs there is nothing to observe, limiting the x axis to 4 second would allow the reader to see more detail in the important part of the results (from 0 to 4 secs).

There is a similar issue with the tracking results. For example in fig 14(a) the y axis starts at 0, however, after 0.5 seconds there is no data below 50 degrees and a large part of the plot area is empty of data. I would suggest cropping the data so it does not include the initial step response from 0 degrees as the step response of the controller has been shown previously.

The tracking experiments have all been performed at very low speed (0.05hz). This is much slower than an exoskeleton would be expected to move, why was such as low frequency chosen. How did the system perform at much higher frequencies?

Reviewer 2 Report

Good comparison of fuzzy sliding surface and PID controller in a well motivated application.

I would have liked more details on how the control parameters in Table 5 were chosen (Gu =5.8 for example) and how you decided what the PID parameters will be for comparison. Are these the parameters that each provide the least error for the respective control method? if so what's the process to find those values? Going into more detail here will give the paper more impact for robotics researchers who seek to apply the results to an actuator.

Line by line the paper was generally well organized with a few minor points

Line (60) Figure 1 not mentioned in the text?

Line (71) What is "the expected tracking performance and robustness," this could be more quantitative

Figure 5 should say "potentiometer"

Line (81) What is meant by the "classification system," is this some kind of state recognition system?

Line (267) defuzzified should be defuzzify

Figures 9 and 10 have a label "PN" that doesnt appear elsewhere --should it be "PM" 

Figure 14B, the big peak at t=0 hides a lot of interesting details on the error signal, but it is consistent with the other plots so maybe should be kept as is

Round 2

Reviewer 1 Report

Thank you for addressing my comments. I am happy for the table to remain if you think this is best.